# The Association of Socioeconomic and Lifestyle Factors with the Oral Health Status in School-Age Children from Pakistan: A Cross-Sectional Study

**DOI:** 10.3390/healthcare11050756

**Published:** 2023-03-04

**Authors:** Maria Moin, Afsheen Maqsood, Muhammad Mohsin Haider, Hajra Asghar, Kulsoom Fatima Rizvi, Abedalrahman Shqaidef, Rania A. Sharif, Ghazala Suleman, Gotam Das, Mohammad Khursheed Alam, Naseer Ahmed

**Affiliations:** 1Department of Community Dentistry, Bahria University Dental College, Karachi 75530, Pakistan; 2Department of Oral Pathology, Bahria University Dental College, Karachi 75530, Pakistan; 3Department of Orthodontics, Faculty of Dentistry, Ajman University, Ajman 346, United Arab Emirates; 4Department of Prosthodontics, College of Dentistry, King Khalid University, Abha 62529, Saudi Arabia; 5Department of Preventive Dentistry, College of Dentistry, Jouf University, Sakaka 72345, Saudi Arabia; 6Center for Transdisciplinary Research (CFTR), Saveetha Dental College, Saveetha Institute of Medical and Technical Sciences, Saveetha University, Chennai 600077, India; 7Department of Public Health, Faculty of Allied Health Sciences, Daffodil International University, Dhaka 1207, Bangladesh; 8Department of Prosthodontics, Altamash Institute of Dental Medicine, Karachi 75500, Pakistan; 9Prosthodontics unit, School of Dental Sciences, Health Campus, Universiti Sains Malaysia, Kota Bharu 16150, Malaysia

**Keywords:** parent education, lifestyle, dental caries, children, dietary habits, oral health

## Abstract

The data on how lifestyle factors of school-going children affect their oral health are not sufficient; therefore, there is a need to analyze the adverse effects of poor lifestyle habits and the role of mothers’ education on oral health. The aim of this study was to analyze the association of socioeconomic and lifestyle factors with the oral health status of school-going children through a structured questionnaire and oral examination. Ninety-five (26.5%) children were from class 1. One hundred eighty-seven (52.1%) mothers were educated while 172 (47.9%) were uneducated. Two hundred seventy-six (76.9%) children had never visited the dentist. The results indicate that dental health behavior is associated with lifestyle factors as well as socio-demographic variables. Parent education and awareness regarding oral health plays a major role in determining the oral health of children.

## 1. Introduction

Dental health is an integral part of general health and well-being. It is not wise to disregard the importance of oral health as a support for general health [1]. Oral hygiene is sustained by keeping the oral cavity free from pathogenic bacteria [2,3]. Oral disease has a multifactorial etiology and needs to be recognized early [4,5]. The emphasis should be on prevention rather than treatment [6]. Historically, oral health has been treated as a separate entity from general health [7]. Poor oral health among children is on the rise, especially in low-income countries [8]. Poor oral health causes impaired nutritional status, which could impede children’s growth and development [9]. Additionally, poor oral health in children can lead to rejection by the general public, which in turn impacts their psychological well-being and their academic performance [8,9]. The data concerning how good and bad lifestyle habits in school children affect oral health are not sufficient. It is also necessary to analyze what role the parents’ educations play in their children’s general health.

In the contemporary era, dietary habits have radically changed, leading to an almost-doubled intake of energy-dense foods (fat, sugar, and salt) and empty-calorie snack foods [10,11]. The daily consumption of different foods in children is a regular routine of life. The dominance of oral health problems is common in children belonging to low socioeconomic backgrounds, as compared to children from prosperous families [11]. Untreated dental caries is the most predominant condition concerning the global burden of diseases [11], and it can have a direct influence on the social life of an individual [12,13]. It may impede the efficiency of a person, resulting in psychosocial distress [12]. In young children, oral health problems can cause significant aesthetic and functional restrictions, making them socially obstinate, anxious, and also lagging behind their companion students in school [12,13]. Pre-primary education is a fundamental part of the Pakistani school system and offers compulsory education for toddlers from 3 to 6 years of age. It also provides childcare and education programs to children under the age of three years [14]. The WHO, as a result, recommends shifting our healthcare system towards a primary care approach, which focuses on the prevention of diseases and promotion of oral health awareness and consequently results in an improved quality of life. Furthermore, it is emphasized that the treatment of disease should be the second approach while keeping prevention as the first line of action [15]. A large portion of the Pakistani population is believed to lack adequate information and support to maintain oral health. The children in particular are believed to have impaired oral health. Therefore, the working hypothesis of this study was that the role of risk indicators related to lifestyle and socioeconomic factors may have an impact on the oral health of children in Pakistan. The aim of this study was to analyze the impact of socioeconomic and lifestyle factors on the oral health status of school-age children through a structured questionnaire and clinical oral examination.

## 2. Materials and Methods

### 2.1. Study Design and Ethical Considerations

A cross-sectional study was conducted according to the standards of the Helsinki Declaration. The study was approved by the ethical and review committee of Altamash Institute of Dental Medicine, Pakistan (AIDM/ERC/09/2021/01). The parents and guardians of the children were informed about the objective and provided with detailed information on the study. The informed consent from parents and guardians was sought out.

### 2.2. Population and Sample Size

According to the report of the Pakistan Bureau of Statistics and Census (2017), the estimated population of children between 3 and 16 years of age was 10,645,272, so the sample size was computed for convenience using EPIDAT 4.0. Considering the population with respect to the age of the children, the sample size was estimated using the following parameters for the assessment of the association between socioeconomic and lifestyle factors and the oral health status: 5% standard of error, 80% power, 95% confidence interval, design effect of 1.4, 10% of non-response, and a prevalence ratio of at least 1.4. The estimated sample size was 359 participants.

To be part of the study, it was obligatory to fulfill the following inclusion criteria: children from 3 to 16 years of age with the consent of a legal guardian; no systemic disability or habit of using fixed devices that hinder the examination; not taken any antibiotics, medicated toothpaste, or mouthwash in the previous 6 months. The data were collected from October to December 2021 from two schools of Karachi, Pakistan: The Noor public school and Four Dots Army Public School.

### 2.3. Selection of Participants

Convenience sampling was employed to select participants. A sample of 425 children was approached first, and the legal guardians of 359 children provided their consent to be a part of the study. The remaining 66 children were excluded from the study because of the refusal of the legal guardians to participate or the uncooperative nature of the children. The study utilized a structured questionnaire followed by an oral examination. The filing of the questionnaire and clinical examination were executed on the same day.

### 2.4. The Study Questionnaire and Oral Examination

The questionnaire was designed to fulfill the aim of the study and consisted of three parts (personal history, lifestyle habits, and dental habits) with a total of 19 items. The questions were asked by the parents or guardians of children from 3 to 6 years of age; whereas, those from 7 to 16 years replied by themselves. The first part of the questionnaire consisted of socio-demographic data such as age, gender, maternal education (matric, inter, graduate, postgraduate, or uneducated), mother and father occupations (professional: doctor, engineer, banker; white-collar worker: school teacher, entrepreneur, nurse, manager; blue-collar worker: clerk, shop keeper, bus driver, carpenter; or unskilled laborer: cleaner, peon), place of residence (urban or semi-urban), self-assessment of oral and general health (average, good, poor, don’t know), the experience of toothache frequency (often, rarely, never), and absenteeism from school (yes, never). The questionnaires also included data on lifestyle habits such as performance of outdoor activities (every day, once a week, twice a week, rarely, never), problems with eating and sleeping (yes, no), tooth brushing habits (once daily, twice daily), frequency of consumption of sugary drinks, frequency of dental visits, and attitude towards oral health, as described in Appendix A.

The oral examination assessed the dentition, periodontal status, gingival bleeding, dental trauma, dental erosions, and oral mucosal lesions based on the methods and criteria of the World Health Organization WHO [16]. Oral examination was performed with a portable dental unit with sterilized instruments under daylight by a single examiner following proper cross-infection control protocol. The dentition status was recorded for both permanent and deciduous dentition by using the Decayed, Missed, Filled Tooth (DMFT) index and decayed, missed, filled, tooth (dmft) index, respectively. The D component represents the number of decayed teeth; the M component represents the number of missing teeth due to caries; the F component represents the number of filled teeth; and the T component represents teeth in permanent dentition. Similarly, the d component represents the number of decayed teeth; the m component represents the number of missing teeth due to caries; the f component represents the number of filled teeth; and the t component represents teeth in deciduous dentition. The periodontal status was assessed with a gingival bleeding score (absence of condition = 0, presence of condition = 1). Enamel fluorosis was assessed by scoring (normal = 0, questionable = 1, very mild = 2, mild = 3, moderate = 4, severe = 5). Dental erosion severity was assessed by scoring (no sign of erosion = 0, enamel lesion = 1, dentinal lesion = 2, pulp involvement = 3), and so was the status of dental trauma (no sign of injury = 0, treated injury = 1, enamel fracture only = 2, enamel & dentine fracture = 3, pulp involvement = 4, missing tooth due to trauma = 5), oral mucosal lesions condition (no abnormal condition = 0, ulceration = 1, acute necrotizing ulcerative gingivitis = 2, candidiasis = 3, abscess = 4, other condition = 8), and location (vermilion border = 0, commissure = 1, lips = 3, sulci = 4, buccal mucosa = 5, floor of the mouth = 6, tongue = 7, hard/soft palate = 8). Alveolar ridge/gingiva and intervention urgency (no treatment = 0, prevention or routine treatment needed = 1, prompt treatment needed = 2, immediate treatment needed = 3, referral = 4) were also recorded for immediate referral and treatment. The flow diagram of the study is shown in Figure 1.

### 2.5. Statistical Analysis

The Statistical Package for the Social Sciences software (IBM, SPSS Statistics, version 25, Chicago, IL, USA) was used for data analysis. The mean values and standard deviations were calculated for the socio-demographic factors, lifestyle, and dental habits along with descriptive analyses such as frequencies and percentages of the given data. The Chi-square test was applied to determine the correlation between dependent and independent variables. Post hoc analysis was performed with a Bonferroni test. The linear regression analysis was performed to predict the effect of age on the independent variables. A *p*-value of ≤0.05 was considered to be statistically significant.

## 3. Results

A total of 359 children—202 males (56.3%) and 157 females (43.7%)—were assessed for the impact of socioeconomic and lifestyle factors on oral health. The age range was a minimum of 3 years to a maximum of 16 years, with a mean age of 8.94 ± 0.173 (Table 1). The majority of the children, 95 (26.5%), were from class 1. Regarding the level of maternal education, 187 (52.1%) of the mothers were educated and 172 (47.9%) were uneducated. Concerning the level of qualification of the mothers, the majority, 122 (34.0%), studied until the 10th grade. Furthermore, 41 (11.4%) of the mothers were working and 318 (88.6%) were housewives. The fathers were mostly from blue-collar occupations, 264 (73.5%). Three hundred eleven (86.6%) children belonged to semi-urban areas. Regarding the mode of feeding in infancy, more than half of children, 193 (53.8%), were breastfed. When questioned for the self-assessment of oral health, 212 (59.1%) children were graded as average. One hundred sixty-three (45.4%) had not experienced a toothache in the previous twelve months. Concerning absenteeism over 12 months, 244 (68.0%) children were present. Of the total participants, 174 (48.5%) were engaging in outdoor activities every day. Most of them had no eating and sleeping problems, 320 (89.1%) and 342 (95.3%), respectively. When asked about their brushing frequency and time, 197 (54.9%) participants reported that they brushed their teeth twice a day with a time duration of 2–3 min. Furthermore, 276 (76.9%) participants had never visited the dentist. Overall, the attitude toward oral health was graded as moderate in 259 (72.1%) of the participants.

Regarding the periodontal health status, 304 (84.7%) of the participants had no periodontal problems. Similarly, 306 (85.2%) had no enamel fluorosis. There were no signs of erosion, dental trauma, and mucosal lesions present in 357 (99.4%), 350 (97.5%), and 359 (100%) participants, respectively. No dental treatment was needed in 216 (60.2%) of the participants. The dentition status analysis of the children was as follows: 64 (17.8%) participants had deciduous teeth in their mouths while a majority, 222 (61.8%), had mixed dentition, and 73 (20.3%) participants had permanent dentition. A total of 1077 teeth were examined in the study. The DMFT score is the sum of the number of teeth decayed, missing, and filled. According to the DMFT index, the number of decayed teeth was 86, the number missing was 60, and there were 53 filled teeth. Table 2 shows that a total of 298 teeth were recorded as decayed, missing, or filled. The total represents the DMFT scores for permanent dentition of all the study participants. Similarly, the statuses of all three parameters for deciduous dentition are presented.

In Table 3, the Chi-square test was applied to the dependent variables (gender, oral health assessment, physical activity, student’s grade, and level of maternal education) and independent variables (DMFT, dmft, periodontal status, brushing time, brushing frequency, sugary drink consumption, absenteeism from school).

The analysis of gender within the independent variables showed a significant association with brushing frequency (*p* = 0.003, x^2^ = 11.692); however, the association with other independent variables was insignificant (*p* ˃ 0.05). Regarding the oral health assessment, the association with the dmft score in primary dentition was significant (*p* = 0.026, x^2^ = 31.389). Furthermore, physical activity was also significantly associated with sugary drink consumption (*p* = 0.040, x^2^ = 16.142) and absenteeism from school (*p* = 0.008, x^2^ = 13.850).

The student’s grade was significantly correlated with the dmft (deciduous decayed, missing, and filled teeth) (*p* = 0.001; x^2^ = 126.308), brushing time (*p* = 0.003; x^2^ = 51.895), and absenteeism from school (*p* = 0.011; x^2^ = 0.011).

Additionally, the level of maternal education was significantly associated with the dmft score in deciduous dentition (*p* = 0.034, x^2^ = 18.076) and brushing time (*p* = 0.006, x^2^ = 12.297).

In post hoc analysis, a significant difference (*p* = 0.008) was noted between the genders and brushing frequencies of the children. Once-daily and twice-daily brushing were prevalent in both males (53.6%) and females (43.7%). Regarding the level of maternal education, a significant difference (*p* = 0.006) was noted between brushing times. Brushing times of more than 2 to 3 min were found with both educated and uneducated mothers (*p* = 0.0009). There was a significant difference (*p* = 0.003) between physical activity and sugary drink intake. Sugary and non-sugary drinks were consumed by the participants with rare physical activity (*p* = 0.0013). Furthermore, a significant difference (*p* = 0.005) was present between physical activity and absenteeism from school. The physically active children were present most in school compared to the absentees. Concerning the self-care of oral health, a significant difference (*p* = 0.008) was noted. The self-assessment was completed by the majority of children, and they were found to be in the good- and average-care categories (*p* = 0.006), Appendix A.

Table 4 describes a regression analysis of one dependent variable (age) with the independent variables (DMFT, dmft, periodontal status, brushing time, brushing frequency, sugary drink consumption, and absenteeism from school) among the participants.

The outcome of the study showed that a moderate correlation between variables exists. The regression model analysis for age to independent variables showed a constant value of 0.418 for R-Squared (R^2^), and the adjusted R-Squared (AR^2^) value was 0.175. However, the independent variables, dmft (*p* = 0.001), periodontal status (*p* = 0.001), brushing time (*p* = 0.007), brushing frequency (*p* = 0.001), and absenteeism from school (*p* = 0.036), showed a significant correlation with the ages of children. DMFT and sugary drink consumption were not associated with the age of the children, indicated by a *p*-value ˃ 0.05.

The age to dmft, periodontal status, brushing time, brushing frequency, and absenteeism from school beta (B) values were statistically significant (B= −0.269, 0.226, −0.135, 3.698, and −0.104), which indicates that these independent variables increased with age, or vice versa.

## 4. Discussion

Socioeconomic and lifestyle factors are broad terms that encompass income, education, community safety, social status, adequate sleep, healthy eating habits, physical and psychological health, and maintaining a proper weight. Their impact on general health is well documented. Studies have shown that these factors affect both sexes [17,18]. As with other age groups, socioeconomic and lifestyle factors also affect the health of children, from preschoolers to university students [19,20]. The impact of lifestyle factors is so vast that a study in 2018 in the United States concluded that people who adopt healthy lifestyles have a lower mortality rate, leading to a greater life expectancy [21]. In Pakistan, studies have been conducted to assess the impact of lifestyle and social factors such as parental education, use of tobacco, areca nuts, and oral health knowledge behaviors on the overall oral health of school children [1,4,12,22,23]. Considering these previous findings, the purpose of the current study was to assess the impact of socioeconomic and lifestyle factors on the oral health status of school-going children in the Pakistani population.

This study involved a sample of 359 school-going children with an age range of 3–16 years. In our study, 26.5% of the children were studying in class I. Almost half (52.1%) of the children’s parents were educated with 34% of the mothers being matric-passed. Level of education plays a key role in determining oral health behaviors. The findings of our study were in coherence with the results of a study from India in 2018 which concluded that parent education is significant in promoting oral health in children [12]. In the current study, the parents’ levels of education overshadowed the fact that a majority of the children were living in a semi-urban area (86.6%) with 73.5% of fathers working in blue-collar occupations. The self-assessment of oral and general health was graded as average for 59.1% of the participants. Almost half of the participants had not felt any sort of toothache leading to absenteeism from school in the previous 12 months. This finding is in line with a study conducted in Saudi Arabia in 2016 involving secondary school students which observed that toothaches were associated with students missing school and lacking in academic performance [24]. A systematic review in 2019 also concluded similar observations, stating that dental caries and tooth pain had a negative association with academic achievement and school absenteeism [25].

Regarding dental habits, 54.9% of children brushed their teeth twice a day with a time duration of 2–3 min. The majority of children (76.9%) reported that they had never visited the dentist because they had not experienced any dental needs. Most of the participants (84.7%) had not experienced any periodontal problems, i.e., fluorosis (85.2), erosion (99.4%) trauma (97.5%), or mucosal lesions (100%). Consequently, 60.2% did not need any dental treatment. All these findings indicate that oral hygiene maintenance was good among the participants.

In the present study, dmft (*p* = 0.001), periodontal status (*p* = 0.001), brushing time (*p* = 0.007), brushing frequency (*p* = 0.001), and absenteeism from school (*p* = 0.036) showed a significant correlation with the age of the children while gender was associated (*p* = 0.003) with brushing frequency. A study conducted in 2019 in Egypt reported similar results in which dmft was positively correlated with the age of the participants. The study involved school children from 3–18 years of age and found that brushing frequency and brushing time were significant factors in the good periodontal status of the children [26]. Similar results were reported by a study conducted in Pakistan in 2015. The authors found that periodontal status was significantly associated with age, gender, occupation, brushing, and other oral hygiene habits. Although the majority belonged to older age groups, 590 (30%) participants belonged to the less-than-20-years-of-age group, which is similar to our study [27]. Maternal education levels were also found to be significantly associated with dmft and brushing in the current study, which is in line with the results of the study carried out by Abbass et al. [26]. A possible explanation for this finding is that when parents are educated, they are expected to be well aware of oral health. They keep a proper check on their child’s brushing and dietary habits, so the chances of cavity formation are consequently lower with decreased dmft scores. Moreover, it is considered that if parents are well aware of the importance of maintaining good oral hygiene, it would result in fewer dental problems, especially dental pain, which is considered one of the causes of children missing school. In the current study, maternal education was found to be associated with a child’s school absenteeism, which is in line with the above explanation. Furthermore, various authors have found tooth pain to play a vital role in a child’s school absenteeism [25]. In a study by Ruff et al., DMFT was found to be significantly correlated with the self-assessment of oral care. Since the majority of the participants in our study reported no pain in the previous 12 months, lack of pain may relate to less reporting of tooth decay. Thus, lack of pain can relate to lower dmft and consequently good self-assessment of oral health.

Physical activity (PA) was also found to correlate with sugary drink consumption and school absenteeism. In the current study, half of the participants were involved in some PA daily. PA is considered one of the healthy lifestyle factors. Relationships in the current study may suggest that more PA for a child will lead to a healthier lifestyle and good attendance and academic performance in school [24,25]. A child’s residence was also found to be significantly correlated with school absenteeism. This may be due to the fact that the majority of the students were living in semi-urban areas (86.6%) and their fathers belonged to working- and manual-labor-class occupations, which may lead to adopting a less-concerned attitude or not making their child’s school attendance a priority. Similar results were reported in a review published in 2013 where the authors found that children living in lower social classes were three times more likely to miss school than those in higher social groups [28]. The residence was also found to be significant with brushing time and sugary drink consumption. This was, again, related to the area of residence, i.e., the semi-urban area, which may indicate a middle–low social class in which people do not have adequate knowledge about maintaining health. This is also backed up by a study on the Portuguese population (2017) in which the author assessed oral hygiene habits in different social classes. They found that by comparing social classes, significantly fewer people from the lower social class brushed their teeth twice a day or were maintaining proper dietary habits [29]. The child’s school grades were also found significant with brushing frequency, DMFT, periodontal status, and absenteeism from school. This may be due to the reason that a child free from disease, i.e., with good general and oral health, will be more likely to be present at school and focus on studies, leading to higher performance and good grades [25]. Educating children as well as adults has been a vital part of preventive dentistry that has evolved over a period of time in order to improve the oral hygiene of individuals. Preventive dentistry begins at the level of primary school and includes visual presentations of brushing and the importance of oral health through various educational videos [30].

This study addressed a broad question assessing the impact of socioeconomic and lifestyle factors on oral health. The overall general and oral health of the students were good. The main findings of this study state that maternal education has a vital role in maintaining a child’s oral hygiene. Most students did not face any dental problems and were regularly at school. Most participants reported general and oral self-assessments as good. The students’ brushing habits were adequate, which was also reflected in lower dmft and periodontal status. Age, gender, residence, physical activity, and student academic performance were found significant with sugary drinks, brushing habits, dmft, periodontal status, and school absenteeism.

Despite its strengths, this study also met some limitations by considering such a broad research question. This study also had a small sample size. Furthermore, it did not approach different schools for data collection from other states, and only the mothers’ education was considered to influence the oral health status of children. Hence, the result might not be generalizable everywhere in Pakistan, and the outcomes may vary. Lastly, it does not consider any query regarding COVID-19, as it is known that the recent pandemic has impacted lifestyle factors throughout the world [31]. This study provides a baseline for researchers who are interested in assessing lifestyle and socioeconomic factors across Pakistan. Furthermore, multi-centered group, comparative cluster-based studies with large sample sizes are required to make such findings generalized. Skill-based learning and school-based health-promoting interventions are recommended to tackle lifestyle factors.

## 5. Conclusions

The results indicate that a number of lifestyle factors influence oral health status; gender and level of parents’ education positively influenced brushing frequency and brushing time. Physical activity influences sugary drink intake and absenteeism; those who rarely perform any physical activity showed absenteeism from school and consumed both sugary and non-sugary drinks. Self-care also showed an association with absenteeism; the children with a sense of good self-care showed zero absenteeism, while it was more frequent in children with average self-care.

The overall results showed that dental health behavior is associated with lifestyle factors as well as socio-demographic variables. Parent education and awareness regarding oral health play a major role in determining the oral health of children.

## Figures and Tables

**Figure 1 healthcare-11-00756-f001:**
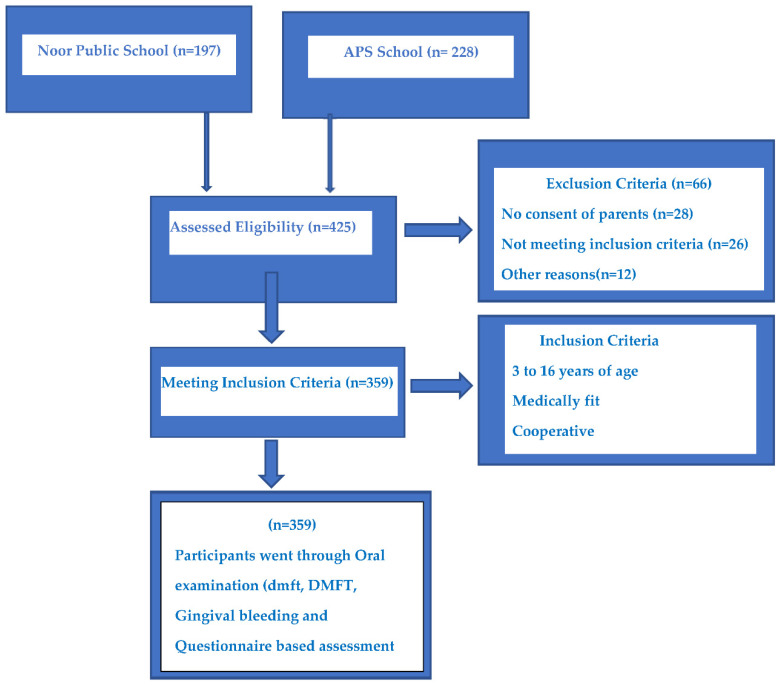
Flow diagram of the study.

**Table 1 healthcare-11-00756-t001:** Distribution of demographic characteristics (*n* = 359).

Characteristics	Number (%)
Gender	
Male	202 (56.3)
Female	157 (43.7)
Level of Maternal Education	
Educated	187 (52.1)
Uneducated	172 (47.9)
Age	Mean and SD
	8.94 ± 0.173

**Table 2 healthcare-11-00756-t002:** Distribution of decayed, missing, and filled teeth in permanent and deciduous dentition (*n* = 359).

DMFT Score	Decayed	Missing	Filled	Total Number of Teeth
Permanent dentition	86	60	53	199
dmft score				
Deciduous dentition	273	299	306	878

DMFT; dental caries index for permanent dentition, dmft; dental caries index for deciduous dentition.

**Table 3 healthcare-11-00756-t003:** Association of dependent and independent variables in the study *(n* = 359).

Dependent Variables	Independent Variables	Chi-Square (x^2^)	df	*p*-Value
Gender	DMFT	9.268	9	0.413
	dmft	11.698	9	0.231
	Periodontal Status	0.208	1	0.648
	Brushing Time	1.024	3	0.796
	Brushing Frequency	11.692	2	0.003 *
	Sugary Drink Consumption	1.331	2	0.514
	Absenteeism from school	13.320	9	0.149
Oral health assessment	DMFT	13.774	10	0.184
	dmft	31.389	18	0.026 *
	Periodontal Status	0.417	2	0.812
	Brushing Time	3.278	6	0.773
	Brushing Frequency	2.709	4	0.608
	Sugary Drink Consumption	0.405	4	0.982
	Absenteeism from school	11.765	2	0.093
Physical activity	DMFT	38.993	36	0.337
	dmft	35.699	36	0.483
	Periodontal Status	6.524	4	0.163
	Brushing Time	14.810	12	0.252
	Brushing Frequency	10.180	8	0.253
	Sugary Drink Consumption	16.142	8	0.040 *
	Absenteeism from school	13.850	4	0.008 *
Student grade	DMFT	52.143	81	0.995
	dmft	126.308	81	0.001 *
	Periodontal Status	14.342	9	0.111
	Brushing Time	51.895	27	0.003 *
	Brushing Frequency	25.004	18	0.125
	Sugary Drink Consumption	16.242	18	0.576
	Absenteeism from school	21.391	9	0.011 *
Level of maternal education	DMFT	5.241	9	0.813
	dmft	18.076	9	0.034 *
	Periodontal Status	0.127	1	0.721
	Brushing Time	12.297	3	0.006 *
	Brushing Frequency	2.251	2	0.325
	Sugary Drink Consumption	0.267	2	0.875
	Absenteeism from school	3.361	1	0.067

* Indicates *p*-value ≤ 0.05 in Chi-square (x^2^) test; DMFT = Decayed, Missed, Filled Tooth for permanent dentition; dmft = decayed, missed, filled, tooth for deciduous dentition; df = degree of freedom; *p* = confidence interval.

**Table 4 healthcare-11-00756-t004:** Linear regression analysis of age and independent variables (*n* = 359).

	Independent Variables	Unstandardized Coefficients		Standardized Coefficients Beta (β)	*t*-Value	*p*-Value	95% Confidence Interval for B		Collinearity	
		B_o_	Std. Error				Lower Bound	Upper Bound	Tolerance	VIF
**Age**	DMFT	−0.049	0.112	−0.023	−0.433	0.665	−0.269	0.172	0.846	1.181
	dmft	−0.410	0.075	−0.269	−5.460	0.001 **	−0.558	−0.263	0.969	1.032
	Periodontal Status	1.735	0.405	0.226	4.283	0.001 **	0.938	2.531	0.845	1.183
	Brushing Time	−0.456	0.168	−0.135	−2.718	0.007 **	−0.785	−0.126	0.950	1.053
	Brushing Frequency	0.999	0.270	0.185	3.698	0.001 **	0.468	1.530	0.944	1.059
	Sugary Drink Consumption	0.038	0.264	0.007	0.144	0.886	−0.482	0.558	0.973	1.028
	Absenteeism from school	−0.612	0.290	−0.104	−2.109	0.036 *	−1.183	−0.041	0.972	1.029

DMFT = decayed, missing, filled teeth permanent; dmft = decayed, missing, filled temporary teeth; B = The rate of change per unit between dependent and independent variables; VIF = Variance inflation factor; t = test of the regression coefficients; * *p* < 0.05; ** *p* < 0.001; B_0_ = unstandardized coefficient, i.e., average estimation of age, gender, level of maternal education, maternal education, self-assessment of oral health, physical activity, residence area, and student grade on DMFT, dmft, periodontal status, brushing time, brushing frequency, sugary drink consumption, and absenteeism.

## Data Availability

The data presented in this study are available upon request from the corresponding author.

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
