# Peer review of "The Association of Socioeconomic and Lifestyle Factors with the Oral Health Status in School-Age Children from Pakistan: A Cross-Sectional Study"

_healthcare, 2023, doi:10.3390/healthcare11050756_

Round 1

Reviewer 1 Report (Previous Reviewer 2)

Dear Colleagues,

This submission is a revised version of an article which I have reviewed at least twice. It is much improved from the original but several issues remain to be addressed. 

While the english is improved with regard to spelling and grammar, there are still abundant instances of phraseology that just sounds bizarre to native speakers. For example the sentence: "Variety of food consumption on diurnal basis in children is normal on regular basis". This is simply one example and to list all would be excessively time consuming. I strongly advise a native english speaker proof reads the manuscript.

For the results of the chi square tests presented in table 3, these may benefit from post-hoc analyses in supplementary data. Especially as there are large degrees of freedom in some of the tests. Thus, significant findings should probably be explored post-hoc or at the very least publish the contingency tables with observed and expected values in supplementary data.

I also note that multiple tests are done and perhaps an adjustment should be explored to control for multiple hypothesis testing. 

Author Response

The comments are addressed in the attached word file

Reviewer 2 Report (New Reviewer)

Respected authors,

the reviewer has to decline recommendation of your article.

- englisch language is not undestandable

- variables/items are not defined (for exampe: what did you consider educated/uneducated? or What is blue collared occupation?

- data in tables are not understandable what information gives - for example student garde - DMFT - Chi 52.143 - df=81 - p=0,995? There is no analytical  interpretation of the data.

- the findings are redundant to many research done in the past years - except for the outcome, that all findings on the influence of socio-economic and lifestyle  factors are contributing to bad oral health also in your country. Much more woud be needed research on solving those problems - how to empower children and families to improve oral health, how to motivate for lifestyle changes. That is, what we can do as researchers in dentistry. You cod use your data to detect families in need and to design helpful interventions and then report if it was successfull. 

Author Response

The comments are addressed in the attached word file

Round 2

Reviewer 2 Report (New Reviewer)

The authors did improve their manuscript.

Nevertheles there are still some concerns:

1. It is unclear, how the sample zise was calculated (nder which assumptions the statistical sample size planning was carried out). Stating that it was done by epidat does not prescribe the calculation process.

2. In Table 2 it is not clear, how many permanent and how many primary teeth were scored with d/m/f (It shold be possible, in a mixed dentition to distinguish between permanent and primary teeth)

3. It is not clearly presented, how the variables gender, OH assessment etc. did influence oral health. In the tabel there is Gender vs. Brushing frequency significant. Okay but these numbers are senseless, because I do not know, if males or females brushed more frequently. The reader might presume what he/she wants. The significancy (p-value) tells us only, that the association of these two variables are not by chance. Please provide a suitable analysis of the data.

4. Statistically, there is a difference between regression and correlation. As stated in the previos concern, authors failed to interpret the statistical data and to state logical findings.

5. Please provide some concrete conclusions. This part is far to vague. No information is given, which particular factors were found to influence oral health.

Author Response

Point to point author team response to reviewer 2 comments

Thank you for reviewing our manuscript. The corrections recommended by the respected reviewer are addressed in different sections of the manuscript. The corrections are highlighted with distinct colors for clarity. The detailed response to the reviewer’s comment is described below:

Reviewer 1 Comments:
Comment 1: It is unclear, how the sample size was calculated (under which assumptions the statistical sample size planning was carried out). Stating that it was done by epidat does not prescribe the calculation process Author’s response: Thank you, the sample size calculation is added in material and methods under heading 2.2 in paragraph 1 at page number 2, line number 88-92. Main document.

Comment 2: In Table 2 it is not clear, how many permanent and how many primary teeth were scored with d/m/f (It should be possible, in a mixed dentition to distinguish between permanent and primary teeth) Author’s response: Thank you, the correction is carried out in table 2, the total number of teeth are added in the last column against permanent, mixed, and deciduous dentitions.

Comment 3: It is not clearly presented, how the variables gender, OH assessment etc. did influence oral health. In the table there is Gender vs. Brushing frequency significant. Okay but these numbers are senseless, because I do not know, if males or females brushed more frequently. The reader might presume what he/she wants. The significance (p-value) tells us only, that the associations of these two variables are not by chance. Please provide a suitable analysis of the data. Author’s response: Thank you, the issue is mentioned in the associated text to table 3. The following information were added:
“In post hoc analysis, a significant difference (p=0.008) was noted between the gender and brushing frequency of the participants, once a day and twice-daily brushing frequency-cy were prevalent in both male(53.6%) and females(43.7%). Regarding the level of maternal education, a significant difference (p=0.006) was noted with brushing time. Brushing time of more than 2 to 3 minutes was found in both educated and uneducated mothers (p=0.0009) While talking about physical activity with sugary drinks intake, there was a significant difference (p=0.003) between them. The sugary and non-sugary drinks were consumed by the participants with rare physical activity (p=0.0013). Furthermore, a significant difference (p=0.005) was present between physical activity and absenteeism from school. The physically active children were present mostly in the school compared to the absentees. In the self-care of oral health, a significant difference (p=0.008) was noted. The self-assessment was done by the majority of children as they were found in good and average care categories (p=0.006). As shown in supplementary table 3.”
Page number 6, line number 245-257. Results section, Main document.

Comment 4: Statistically, there is a difference between regression and correlation. As stated in the previous concern, authors failed to interpret the statistical data and to state logical findings. Author’s response: Thank you, the association of dependent and independent variables in the study was performed with Chi-square test. Post hoc analysis was done with Bonferroni test as per reviewer suggestion to clarify on certain variables, this was done to reduce the chances of obtaining false-positive results, when multiple pair wise tests are performed with a set p-value. (Supplementary table 1).
Furthermore, because the age is a quantitative variable of the study, for this reason to analyze it with multiple variables, we adopted the use of regression analysis mainly to predict the effect of age on independent variables.
This statistical approach was adopted after extensive discussion of authors’ team with statistician, the change in results section of the paper is evident in different submission copies of the manuscript.

Comment 5: Please provide some concrete conclusions. This part is far too vague. No information is given, which particular factors were found to influence oral health. Author’s response: Thank you, revised and the factors influencing oral health is added in conclusion section, page number 11 main document.

This manuscript is a resubmission of an earlier submission. The following is a list of the peer review reports and author responses from that submission.

Round 1

Reviewer 1 Report

Dear authors, 

Please find my suggestions and improvements to be reflected in your manuscript. 

Author Response

Thank you for reviewing our manuscript with great interest, we admire the suggestions, they will help in the improvement of our paper. The point-to-point author response file is attached for the respected reviewer to evaluate.

Reviewer 2 Report

Dear Authors,

I have read your paper entitled "" with interest. I feel the work does has merit and is interesting although may be better viewed and presented from a different perspective.

I have several comments that must be addressed prior to acceptance for publication.

1. The English is not brilliant. Not only are there typos and grammatical errors but also very odd phraseology is used throughout. The tone is often overly informal as well. The work must be proof read by a native English speaker to correct this. There are too many occurrences to list. 

2. The methods must be expanded. Please include the full questionnaire (in supplemental information) and remove figure 2, it doesn't add very much. Please elaborate on the nature of how the oral exam was conducted and recorded in more detail rather than just linking to the WHO citation.

3. The results should be presented in a clearer way. The table is excessive with little actually useful information. Perhaps a correlation matrix to show the strength of the various relations could be employed.

4. The results are not massively novel, at least in how you choose to explore them. The relationship between oral health and socioeconomic status has been well established and known for decades. This alone doesn't mean your study is invalid of course, as it confirms and adds to the literature. To me, the main new finding, or at least more interesting, is between the students grades and their oral health status and behaviour. I do not know how much this has been explored previously, but to me it is interesting and merits further analysis. Perhaps it is a useful marker of personality traits such as conscientiousness.

I feel the above are necessary to address prior to acceptance of the manuscript.

Author Response

(The authors gave the same response as above.)

Reviewer 3 Report

The major problem with this article is that the statistical methods used are not appropriate.

Using age gender, educational attainment etc as a dependent variable doesn’t make sense. Are the authors trying to predict age, gender, education etc using lifestyle and other factors?

Linear regression models (LM) are used to model continuous variable. How came authors used LM for categorical variables such as gender, education and even age categories (as mentioned in the beginning that the age is divide into three categories).

There should be a clear objectives/aims of what authors are trying to predict and what are the predictors.

Appropriate regression models should be used based on the characteristics of the outcome variable (dependent variable) that fulfil the assumption of models with the help of experienced/accredited statistician.

No discussion about the scale and unit of lifestyle and other factors. There should be a table of descriptive measures of all the variables considered in this study (independent and dependent). Also how did they measure these factors such as self-assessment of oral health care, physical activity, residential area; and also, DMFT/dmft (I don’t know what these are, and are they different, one in lower case and other in upper case), periodontal status, sugary drink consumption etc.

Many terms have been used in Table 2 without any explanation or discussion.

Author Response

(The authors gave the same response as above.)

Round 2

Reviewer 2 Report

I feel the manuscript has been improved 

Author Response

Comment 1:

I feel the manuscript has been improved

Author’s response:

Thank you, we are grateful to the reviewer.

Reviewer 3 Report

No comments

Author Response

Point-to-point author team response to reviewer comments

Thank you for reviewing our manuscript. The corrections recommended by the respected reviewers have been addressed in different sections of the manuscript. The track change is on in the manuscript. The detailed response to the reviewer’s comment is described below:

Reviewer 3 comments

Comment :

The major problem with this article is that the statistical methods used are not appropriate.

Author’s response:

Thank you for the valuable comments, the author’s team is grateful, and we revised the results extensively, after a 2nd opinion from a statistician, the categorical variables in the study were analyzed with the Chi-Square test. Table 3 is added with relevant text in the results section. This was carried out after careful inspection of data distribution.

The binary logistic regression possibility was checked by the author’s team, but it was not applicable as per the data.

The linear regression analysis has been confined to the age with independent variables analysis. Since the age is given as a mean value. For clarification, the SPSS data is attached as a supplementary in the system. Linear regression analysis can be successfully applied to age.

Table number 2 is added for demographic variables distribution in the table.

Moreover, the discussion section has been modified according to the changes brought in the results of the study
